# Effects of Ground Slopes on Erector Spinae Muscle Activities and Characteristics of Golf Swing

**DOI:** 10.3390/ijerph20021176

**Published:** 2023-01-09

**Authors:** Bairan Li, Junsig Wang, Chaojie Wu, Zhe Hu, Jiaying Li, Sang-Cheul Nam, Ze Zhang, Jae-Kyun Ryu, Youngsuk Kim

**Affiliations:** 1Department of Physical Education, Putian University, Putian 351100, China; 2Department of Physical Education, Jeonbuk National University, Jeonju 54896, Jeollabuk-do, Republic of Korea; 3Department of Sports Medicine, KyungHee University, Youngin 17104, Gyeonggi-do, Republic of Korea; 4College of Physical Education, Pingdingshan University, Pingdingshan 467000, China; 5Department of Coaching, KyungHee University, Youngin 17104, Gyeonggi-do, Republic of Korea

**Keywords:** golf swing, slopes, erector spinae, injury

## Abstract

(1) Background: ‘Slope’ refers to the position faced by golfers on the course. Research on the recruitment strategies of thoracolumbar erector spinae during golf swings on different slopes may help us to understand some underlying mechanisms of lower back pain. (2) Purpose: The purpose of the present study is to assess electromyography (EMG) patterns of the erector spinae muscles (ES) and the kinematics of the trunk and swing parameters while performing golf swings on three different ground slopes: (1) no slope where the ball is level with the feet (BLF), (2) a slope where the ball is above the feet (BAF), and (3) a slope where the ball is below the feet (BBF). Furthermore, the present study evaluates the effect of slope on the kinematics of the trunk, the X-factor angle, and the hitting parameters. (3) Methods: Eight right-handed recreational male golfers completed five swings using a seven-iron for each ground slope. Surface electromyograms from the left and right sides of the ES thoracolumbar region (T8 and L3 on the spinous process side) were evaluated. Each golf swing was divided into five phases. Kinematics of the shoulder, trunk, and spine were evaluated, and the ball speed, swing speed, carry, smash factor, launch angle, and apex were measured using Caddie SC300. (3) Results: The muscle activity of the BAF and BBF slopes was significantly lower than that of the BLF slope during the early follow-through phase of the thoracic ES on the lead side (i.e., left side) and during the acceleration and early follow-through phases of the lumbar ES on the lead side. The lead and trail side (i.e., right side) lumbar ES were more active during acceleration than the thoracic ES. Additionally, the trends of the lead and trail sides of the thoracolumbar regions on the three slopes were found to be the same across the five phases. Trunk angle and X-factor angles had no significant differences in address, top of backswing, or ball impact. The maximum separation angles of the X-factor appeared in the early phase of the downswing for all the three slopes. Regarding smash factor and launch angle, there were no significant differences between the three slopes. The ball speed, swing speed, carry, and apex were higher on BLF than on BAF and BBF slopes. (4) Conclusion: The findings suggest that amateur golfers face different slopes with altered muscle recruitment strategies. Specifically, during the acceleration phase of the golf swing, the BAF and the BBF slopes, compared with the BLF slope, significantly underactivated the lead side thoracolumbar erector spinae muscles, thereby increasing the risk of back injury. Changes in muscle activity during critical periods may affect neuromuscular deficits in high-handicap players and may have implications for the understanding and development of golf-related lower back pain. In addition, the X-factor angle was not affected by the slope, however, it can be found that the hitting parameters on the BLF slope are more dominant than on the other slopes.

## 1. Introduction

A golf course is generally designed according to the landform on which it is situated. The designer controls the difficulty of the golf course by taking advantage of environmental factors such as trees, sandy areas (which may become bunkers), water, grassy areas, and the slope of the ground [1]. The slope is one of the factors that make the golf game difficult and exciting by changing the height of the ball with respect to the position of the foot. According to Peter et al. [2], approximately 80% of swings on most golf courses are conducted on a slope of plus or minus 10°, which may be an uphill slope, a downhill slope, a slope on which the ball was above the feet (BAF slope), or a slope on which the ball is below the feet (BBF slope) [3,4]. According to previous research on slope, a change of ground slope may lead to a series of changes including weight transfer mode and body coordination, and is one of the main factors that directly affect the ideal golf swing. [3,5,6].

The slope in a game of golf may affect the golfer’s posture at the address. To hit the ball with stability, the golfer should adjust the center line of the body to adapt to the change in the slope [5]. The trunk should lean forward on a BBF slope [7,8], whereas the forward angle of the torso should be relatively upright on an BAF slope [9]. When a motion is introduced in one plane, it will cause a change in the motion of the other two planes [10]. Accordingly, changes in the forward angle of the trunk when facing various ground slopes may have an effect on the range of rotation of the torso during a golf swing [10]. The X-factor is the degree of rotation of the shoulders relative to the hips during a golf swing [11,12,13,14]. Studies by some scholars have demonstrated that a relatively large X-factor at the top of the backswing may help improve hitting performances such as club head speed and distance [11,15,16]. These previous results may indicate that changes in the ground slope affect the X-factor and, thus, the hitting performance. In addition, it is worth noting that the research results of Cohn and Lindsay show that the generation of an excessive X-factor angle will cause excessive torsional load on the lower back and may exert pressure on surrounding soft tissues [17,18].

The golf swing is a complex movement that requires coordinated muscle activity to effectively transmit the force and momentum generated by the golf club [19]. Previous studies have emphasized the important role of the trunk in the kinetic chain of a golf swing [20], not only as part of the force generation, but also as a transmission sequence. As a result, during a golf swing, especially in the downswing phase, excessive lateral flexion and rotational motion of the trunk creates a large load on the spine and back musculature [21]. For example, the highest mean thoracic and lumbar ES activation levels were displayed in the forward swing and acceleration phases of the golf swing for all clubs tested [22,23], and the compressive force on the L4-L5 vertebrae reached 6.5 times to more than 8 times the body weight immediately after impact during the golf swing, compromising the stability of the spine [24,25].

Spinal stability is highly dependent on the spinal load and posture [26] and the task requirements [27]. The golf swing is considered to be a complex and asymmetrical movement that causes unbalanced loads on the musculature of the body, especially when the golfer is not in the correct plane [25]. When facing various ground slopes, golfers change their postures in the sagittal plane. The erector spinae (ES) group serves as the main extensor group of the back and helps stabilize the spine because of its attachments to the pelvis, vertebrae, and ribs [28]. A decrease in trunk flexion/lateral flexion during the downswing is associated with a decrease in left-side ES activity [29]. Playing golf in an upright position closer to the ball prevents back pain because it should reduce vertebral flexion, thereby reducing anteroposterior shear forces on the spine [30]. A previous study [31], using EMG, demonstrated an exponential increase in trunk muscle activation and subsequent torque during rotation in a flexed position compared to a more upright posture; this led to the conclusion that such rapid increases in tissue stress with asymmetric axial rotation could increase injury potential.

There is currently no literature that specifically compares the EMG activity of the lower back muscles across various slopes. Evaluating the EMG activity of these muscles during golf swings on various ground slopes would help us to understand some of the underlying mechanisms of lower back pain or injuries in golf. The purpose of the present study is to compare EMG activities and patterns for the thoracolumbar region of the ES and their kinematics on three different ground slopes. The study hypothesized that the thoracolumbar ES would show higher muscle activation values on BLF and BBF slopes than on BAF slopes; that is to say, the stress on the back would bear a greater load on BLF and BBF slopes than on BAF slopes; in addition, another goal of the present study is to evaluate the effect of slope on the kinematics of the trunk, X-factor, and impact performance parameters (i.e., ball speed, swing speed, carry, smash factor, launch angle, and apex). We hypothesize that the X-factor separation angle produced under the BLF slope is greater than that of the BAF and BBF slopes, and the hitting parameters are also more advantageous.

## 2. Methods

### 2.1. Participants

Eight healthy right-handed male recreational golfers with no history of musculoskeletal injury participated in the study. The study was approved by the Ethics Committee of Jeonbuk University (JBNU2022-04-008-001). The participants had no history of orthopedic problems that might limit their golf swings. Written informed consent was obtained from each participant before the experiment. All participants completed a short questionnaire about their personal information (e.g., height, weight, handicap) before participating in the study. The physical characteristics of the participants are shown in Table 1.

### 2.2. Experimental Procedure and Apparatus

Participants were asked to perform 15 hits at a distance of 3 m from the net (specification: length 5 m × width 2 m × height 5 m) (Figure 1). Five strikes were carried out for each slope, namely, the BLF slope (0°), BAF slope (+10°), and BBF slope (−10°), and the three best trials for each slope were selected for data analysis. Participants hit a regular golf ball (Titleist Pro V1X series) from a golf mat made of artificial turf using their own seven-irons, gloves, and shoes. 

EMG signals during golf swings were collected using a Trigno Wireless sensor (Delsys, Natick, MA, USA). Partial skin preparation (cleaning and shaving) was performed to reduce the impedance of the interface between the skin and the electrodes. Following the skin preparation, EMG electrodes were placed 30 mm lateral to the spinous processes of the eighth thoracic (T8) [22,32] and third lumbar (L3) [23] vertebrae of the left or right ES (L-ES or R-ES).

Before EMG recordings were made, participants performed a 6-min golf-specific warm-up, including their golf swings. After the warm-up was completed, the EMG data on the subjects’ maximum voluntary contractions (MVCs) were collected as a reference for the standardized procedure [33]. For the MVC examinations, the subjects lay prone on a bed lifting their back and legs vigorously [34]. Each subject performed two maximum contractions for 5 s, and the recorded maximum muscle activation value was considered to be the MVC for each trial.

During the experiment, Caddie SC300GPS (Voice Caddie, Seoul, Republic of Korea) was used to record the ball speed (the velocity of the ball leaving the clubface), swing speed (referred to as clubhead speed, this refers to how fast the clubhead is travelling at impact), carry (the distance the ball travels through the air), smash factor (the ball speed divided by the clubhead speed: the higher the smash factor the better the energy transfer), launch angle (measured as an angle up from the ground, reflecting the initial trajectory of the ball), and apex (the maximum height of the golf ball’s trajectory). The SC300 was placed 1 m behind the golf ball facing the golfer’s target line.

### 2.3. Data Collection and Analysis

Raw EMG signals were processed by a 16-bit analog-to-digital converter; EMG data were sampled at 1440 Hz and digitally filtered (25 to 450 Hz); and root mean square (RMS) values were calculated. All EMG analyses were performed using EMGworks (Delsys). Three-dimensional (3D) motion data were measured using the OptiTrack Motion Capture System (NaturalPoint, Inc., Corvallis, OR, USA), which includes 13 high-speed cameras operating at 360 Hz. Fifty-seven reflective markers were placed on the participants (Figure 2), and static models were created for the subjects before the experiment to facilitate later modeling in the analysis software. After the static model was built, three reflective markers were placed on the seven-iron to build the club model. The 3D motion analysis system was synchronized with the EMG data via Motive (NaturalPoint, Inc., Corvallis, OR, USA) so that the synchronization of specific points in the EMG data could be identified by visual observation of the kinematic data. After each experiment, the 3D motion data were transferred to Visual3D software (C-Motion, Inc., Germantown, MD, USA) for subsequent data analysis of such features such as the joint angle.

For each swing, there were six time points: address (AD), top of backswing (TBS), middle of downswing (MDS), ball impact (BI), middle of follow-through (MFT), and finish of follow-through (FFT) (Figure 3). According to the six time points, each golf swing was divided into the following five phases for analysis: (1) backswing phase, from the address of the ball to the top of the backswing; (2) forward swing phase, from the top of the backswing to the middle of the downswing; (3) acceleration phase, from the middle of the downswing to ball impact; (4) early follow-through phase, from ball impact to the middle of follow-through; and (5) late follow-through phase, from the middle of follow-through to the finish of follow-through. The following angles were calculated for analysis: (1) the forward angle of the trunk (defined as the angle of the trunk in the sagittal plane, calculated by thorax relative to Lab in Visual3D); (2) the roll angle of the trunk (defined as the angle of the trunk in the frontal plane, calculated by thorax relative to Lab in Visual3D); and (3) the shoulder-hip separation angle (defined as the X-factor angle, using the left/right clavicular-acromial joints (LCAJ and RCAJ) and left/right iliac anterior spines (LIAS and RIAS), with the angles shown in the transverse plane.

### 2.4. Statistical Analysis

Data were processed in Prism Statistical Software 9.0 (GraphPad Corporation, San Diego, CA, USA). Descriptive statistics were reported as the mean ± SD of the percentage of MVC. Normality was assessed by the Shapiro–Wilk test. A two-way repeated measures ANOVA (slope × phase) was performed. All performance variables using Caddie SC300GPS were subjected to a one-way ANOVA (slope) for statistical significance. The alpha level was set at 0.05 and then adjusted by the Bonferroni correction according to the number of statistical comparisons. Pairwise comparisons were performed with the Bonferroni test.

## 3. Results

### 3.1. Comparison between the Slopes

The activation patterns were the same across slopes, showing an inverted U-shaped trend. However, EMG mean values generally tended to be higher in swings with BLF slopes, especially in the lead side ES thoracolumbar region (e.g., in the acceleration phase, muscle activity on the lead-side lumbar ES showed BLF 49% MVC > BBF45% MVC, BAF 45% MVC) (Figure 4) (Table 2 and Table 3).

### 3.2. Comparison between the Phases

During the backswing phase, the lead and trail sides of the ES had an activation percentage of 8.1% to 26.1% MVC. From the backswing phase to the forward swing phase, the level of muscle activation of the ES in the thoracic and lumbar regions was not significant (*p* > 0.05) (Figure 4) (Table 2 and Table 3).

During the forward swing phase of the golf swing, the lead and trail side ES activation rates in the thoracolumbar region were 13.8% to 30.8% MVC. When comparing the forward swing phases, the level of muscle activation of the ES in the thoracic and lumbar regions was not significant (*p* > 0.05) (Figure 4) (Table 2 and Table 3).

During the acceleration phase of the golf swing, the activation of the thoracolumbar region ES was 29.4% to 57.2% MVC. On the lead side of the lumbar ES L3, the activation percentage was significantly different on the BLF than on the BBF and BAF slopes, and on the trail side of L3, there was a significant difference in percentages on the BBF and BAF slopes. There was no significant difference between the lead side and trail side of the thoracic ES T8 (*p* > 0.05) (Figure 4) (Table 2 and Table 3).

During the early follow-through phase of the golf swing, the thoracolumbar region of the ES had activation percentages of 24.5% to 42.9% MVC. There were significant differences for the lead T8 and lead L3 on the BLF, BBF, and BAF slopes and for the lead L3 on the BBF and BAF slopes (Figure 4) (Table 2 and Table 3).

During the late follow-through phase of the golf swing, the thoracolumbar region of the ES had activation percentages of 16.5% to 40.6% MVC. The percentages for the lead T8 were significantly different on the BLF than on the BBF and BAF slopes. There were also significant differences in percentages for the trail L3 on the BLF and on the BAF slope. There was no significant difference in percentages between the trail T8 and the lead L3 (Figure 4) (Table 2 and Table 3).

### 3.3. Comparison of Trunk Angle and X-Factor

The sagittal plane trunk angle of the BLF, BBF, and BAF slopes (i.e., forward flexion) made no significant difference in the AD, TBS, MDS, BI, or MFT events, however, there was a significant difference in the FFT event (Figure 5) (Table 4).

The frontal plane trunk angle on the BLF, BBF, and BAF slopes was associated with no significant differences at each time (Figure 5) (Table 5).

The X-factor angles on the BLF, BBF, and BAF slopes had no significant differences in AD, TBS, or BI. In addition, it was found that the maximum separation angles on the three slopes were not significant, and their maximum separation angles all appeared in the early phase of the downswing (Figure 6) (Table 6).

### 3.4. Comparison of Hitting Performance Variables

The performance results measured by the Caddie SC300 launch monitor show that there was no significant difference in smash factor and launch angle between the three slopes (*p* > 0.05). However, there were significant differences in ball speed, swing speed, carry, and apex. Post-hoc showed that ball speed, swing speed, carry, and apex in BLF were better than the other two slopes. (See Table 7).

## 4. Discussion

The purpose of the present study was to evaluate EMG activity in the ES of the thoracolumbar region during seven-iron shots on different slopes. We hypothesized that the thoracolumbar ES would show higher muscle activation values on BLF and BBF slopes than on BAF slopes. Furthermore, this study also evaluated the effect of slope on the kinematics of the trunk, the X-factor angle, and the hitting parameters; we hypothesized that the X-factor separation angle produced on the BLF slope is greater than that of the BAF and BBF slopes, and that the hitting parameters are also more advantageous. One main finding was that the muscle activity of the BAF and BBF slopes was significantly lower than that of the BLF slope during the early follow-through phase of the thoracic ES on the lead side and during the acceleration and early follow-through phases of the lumbar ES on the lead side. The second key finding was that the lead and trail side lumbar ES were more active during acceleration than the thoracic ES. Moreover, the trends of lead and trail sides of the thoracolumbar regions on the three slopes were found to be the same across the five phases.

In this study, we hypothesized that the thoracolumbar ES would show higher muscle activation values on the BLF and BBF slopes than on the BAF slope. According to previous findings by Marras et al., ES activity significantly increases when the trunk is twisted in flexion relative to upright trunk twisting [35]. However, the results of this study only showed that the BAF slope was smaller than that of BLF and BBF in the lumbar ES muscle activity on the lead side during the early follow-through phase, which may indicate that a BAF slope inclined towards an upright trunk posture is more conducive to the protection of the lower back after hitting the ball.

A key finding of this study is that during the acceleration and early follow-through phases on the BAF and BBF slopes, the lead-side ES region of the thoracolumbar showed significantly lower muscle activity than on the BLF slope, especially in the early follow-through phase. Reduced back muscle activity seems to reduce the load on the spine and the potential for a lower back injury, however, since this was viewed in the acceleration phase of the golf swing, the spine was undergoing high shear and compression forces [22], and, if muscle activation was insufficient, damage to the spine could result. Without adequate support from the surrounding musculature, the lumbar spine is inherently unstable [36], meaning that reduced muscle activity may increase the risk of back injury at the expense of trunk stability. Therefore, the reduction in ES activity caused by golf swings on uneven ground may be related to a reduced ability to protect the lower back in key parts of the swing when the spine is under high stress [36]. In addition, it is interesting that the lead-side thoracic ES in this study was more active in all three slopes in the early follow-through phase than in the acceleration phase. According to the previous literature by McHardy et al. [37], it was considered that the rotation of the trunk will decelerate in the early follow-through phase, and the activity of the trunk muscle groups will be reduced in this phase relative to the acceleration phase. However, the results of our study show that the lead-side ES activity of the thoracic increases, which may be because the lead side thoracic ES does not fully release the force after the amateur golfer hits the ball, and because the left elbow was too pulled out, which is the ‘chicken wing’ phenomenon that we usually see in an amateur golfer’s swing. This situation is likely to lead to back injuries or shoulder dysfunction [38].

It is clear that the activation of the lumbar ES on the lead and trail sides in the acceleration phase is greater than that of the thoracic ES. This may indicate that the lower lumbar region becomes the main rotator muscle group in the acceleration phase. If one studies the anatomy of the lumbar spine, one sees that the spinal joints, called facets, are oriented to allow flexion and extension, not rotation [39]. If the lower back is forced to become the main rotator muscle group at this point, it would likely be due to insufficient hip and thoracic mobility in amateur golfers, which would likely increase the risk of lumbar spine injury [40].

In addition, we found that the trends of the lead and trail sides of the thoracolumbar region on the three slopes were the same in the five phases. There was a gradual strengthening at the beginning, a peak in the acceleration phase, and then a gradual weakening; similar trends were seen in the lead T8, but unlike in other muscles, the peaks of all three slopes occurred during the early follow-through phase. In the backswing phase, there was no strong muscle activity, so this phase would not be likely to cause damage to the back; when the downswing began, we found that the muscle activity increased. The ES were most active during the acceleration phase due to the rapid rotation of the player’s body and trunk; this is logical because there is high compression of the site during this phase. With the completion of subsequent movements, the level of ES thoracolumbar activation decreased; this is a result affirmed by Graeme G et al. [22]: the body gradually relaxed because of the full release of the force.

In this study, the ES activation of the lead and trail side thoracolumbar region in the backswing and forward swing phases were around 20% MVC, and there was no significant difference. In the early follow-through phase, the lead side thoracic ES and trail side lumbar ES activity found significant differences, but since the muscle activity range of the ES at this stage was basically around 20% MVC, the risk of causing lumbar injury was basically low, and there was no statistical significance.

Although there was no difference in trunk angle before the late follow-through phase on the three slopes, it could be seen that the recreational golfers seemed to change their ready stance as the slope changed (sagittal, BLF 38.92°, BBF 39.73°, BAF 38.36°), findings that are confirmed by Carr et al. [5]; the center line of the body adjusts with the change of slope. Regarding the X-factor, Choi et al. [13] found that the angle of the X-factor on the BAF was much smaller than that on the BLF. Jang found similar results [41]. Contrary to the previous findings of Choi et al. [13], we found that the X-factor did not have significant differences on different slopes. In this study, at the time of the AD and BI, the angle on the BLF was larger than on the BAF and BBF slopes; however, at the time of the TBS, the BLF (71.15°) was smaller than the BBF slope (74.09°) and BAF slope (71.98°). In addition, it was found that the X-factor maximum on the BLF (76.76°) was smaller than on the BBF slope (77.97°). This is probably because recreational golfers raise their arms too much during the backswing in order not to lose distance due to an abnormal stance [11]. This increases the X-factor due to a reduced rotation of the body, produces too much side bending of the body, and impairs the overall stability of the swing, directly leading to a loss of distance, and it may even cause injury.

Regarding the hitting performance parameters, the results of this study show that ball speed, swing speed, carry and apex are higher on the BLF than on the other two slopes (*p* ≤ 0.05) (Table 2). There are differences in swing speed, which may be related to the player’s experience and handicap. High handicap players mainly practice in flat conditions and lack experience hitting balls on different slope [42]. Therefore, the differences in ball speed, swing speed, carry and apex in this study may be related to the experience and obstacles of the golfer [42]. Generally, when facing the BBF slope, it is easy to pull up or tiptoe when hitting the ball, which affects the stability of the hit. This may also be the reason why ball speed, swing speed, carry and apex cannot reach the level of BLF. Additionally, according to the coaching literature [43], in the BAF slope, because the distance between the ball and the body is shortened, in order to avoid hitting the ball deeply (the club head touches the ground in advance), golfers generally choose a shorter grip (hand slightly under handle). The shortening of the length shortens the moment arm acting on the ball when hitting the ball. Under the same force, the energy acting on the ball may decrease, resulting in a decrease in the speed of the ball leaving the face (ball speed), which also affects the flying height of the ball (apex) and the hitting distance (carry). Therefore, most coaches suggest to choose a longer club (e.g., a six-iron rather than a seven-iron, which is a longer club with less loft on the clubface) to cope with the BAF slope in order not to lose distance [43,44].

Interestingly, although there were significant differences between ball speed and swing speed in the present study, the smash factor (smash factor = ball speed/swing speed) shows no significant difference. The reason was most likely because both ball speed and swing speed decreased in the BBF and BAF slopes. Although the smash factor results showed no significant difference, the simultaneous reduction in ball speed and swing speed may have an impact on the golfer’s control of the distance. Unlike studies by Blenkinsop et al. [3] and Hiley et al. [43], the launch angle was not affected by the slope in the present study. This would be because the previous studies evaluated swings in the uphill and downhill slopes. When facing the uphill slope, it is recommended that the player try to keep the posture perpendicular to the ground for the stability of the shot [43]. Meanwhile, players generally respond to uphill slopes by increasing the attack angle to hit a high ball, with the launch angle being the sum of the attack angle and dynamic loft angle [45]. The increase in the attack angle may lead to an increase in the launch angle. However, in the present study, the slopes were the BBF and BAF, and the inclination angle of the rod surface did not change much. This should explain no significant effect on launch angle.

The results of this study may help clinicians and golf coaches better understand the patterns of the ES in the thoracolumbar region when a player is on different types of slopes. This knowledge could help to reduce the possibilities of injury to recreational golfers. The current study shows that the ES of the lumbar spine have high activation levels. Insufficient trunk strength and stability can make recreational golfers more vulnerable to injuries when faced with abnormal stances.

The shortcomings of this paper are that the study was carried out in a closed field and that the hitting performance parameters were all measured by Caddie SC300; these aspects could not provide the same quality of data as the real field. In addition, we selected a small number of participants and all of them were recreational players with relatively low levels of expertise (i.e., handicaps of 16 to 21). In the future, we should recruit more participants and conduct in-depth research on amateur players and professional players at a higher level and compare the two groups. The data in this study were only for right-handed male amateur golfers, and the findings may not apply to left-handed or female golfers.

## 5. Conclusions

The findings suggest that amateur golfers face different slopes with altered muscle recruitment strategies. Specifically, during the acceleration phase of the golf swing, the BAF and the BBF slopes, compared with the BLF slope, significantly underactivated the lead sides thoracolumbar erector spinae muscles, thereby increasing the risk of back injury. Changes in muscle activity during critical periods may affect neuromuscular deficits in high-handicap players and may have implications for the understanding and development of golf-related lower back pain. In addition, the X-factor angle was not affected by the slope, but it can be found that the hitting parameters on the BLF slope are more dominant than on other slopes.

## Figures and Tables

**Figure 1 ijerph-20-01176-f001:**
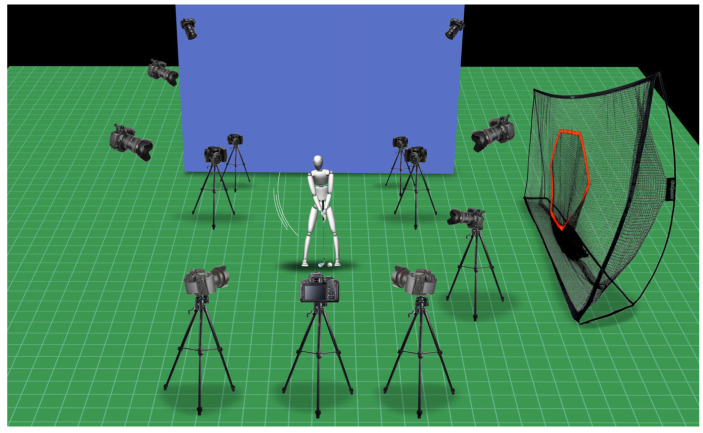
Experimental set-up. EMG and motion data were synchronized by Motive. Caddie SC300GPS was used to measure swing performance.

**Figure 2 ijerph-20-01176-f002:**
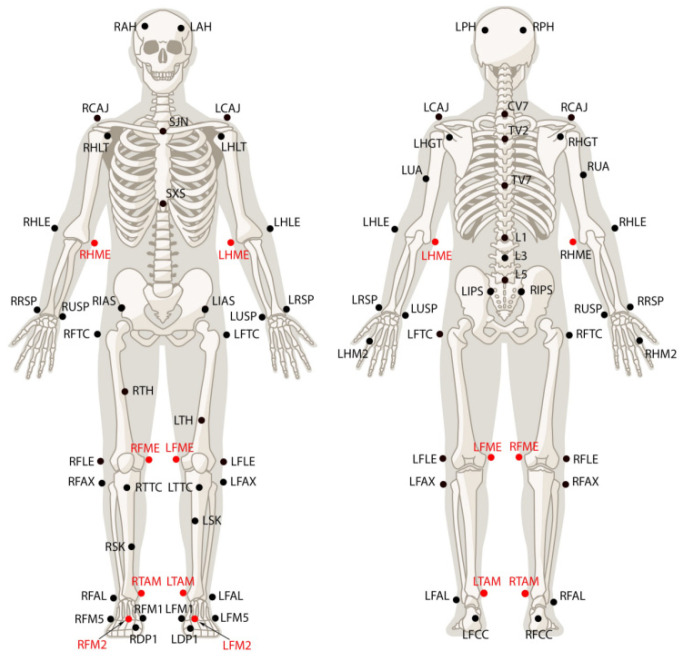
Reflective marker attachment positions. 57 markers were placed on landmarks to build a model predefined by Motive (NaturalPoint Inc., Corvallis, OR, USA).

**Figure 3 ijerph-20-01176-f003:**
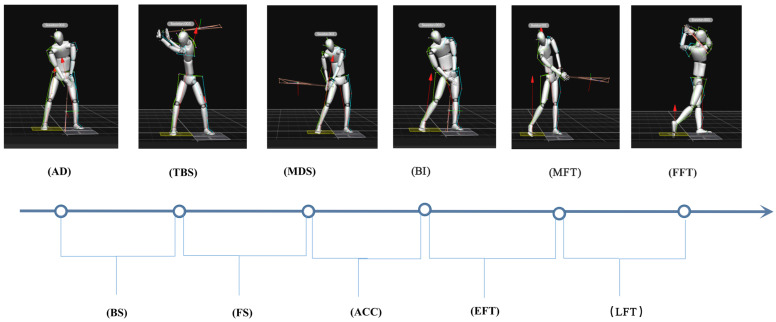
Phases of a Golf Swing. The golf swing was divided into five phases; backswing (BS), forward swing (FS), acceleration (ACC), early follow-through (EFT), and late follow-through (LFT).

**Figure 4 ijerph-20-01176-f004:**
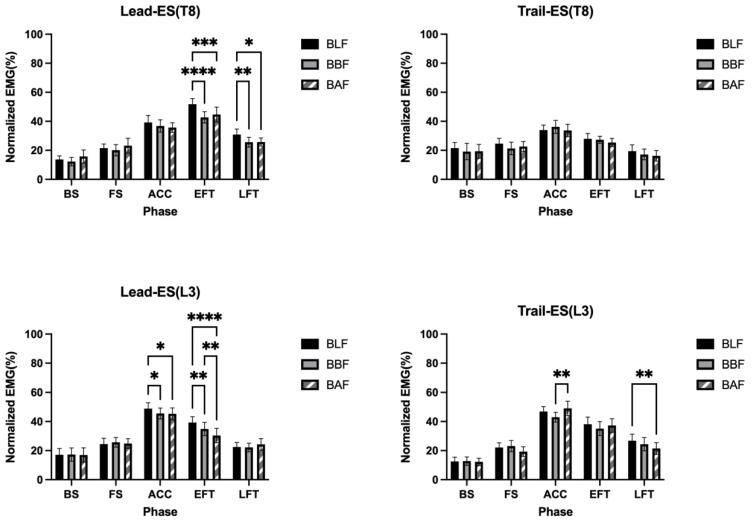
Muscle activities for the lead-ES (T8), trail-ES (T8), lead-ES (L3), and trail-ES (L3) throughout the 5 phases. (BS) backswing, (FS) forward swing, (ACC) acceleration, (EFT) early follow-through and (LFT) late follow-through phase. Comparisons during trials on no slope (BLF), Ball below Feet (BBF) and Ball above Feet (BAF). * *p* < 0.05, ** *p* < 0.01, *** *p* < 0.001, **** *p* < 0.0001.

**Figure 5 ijerph-20-01176-f005:**
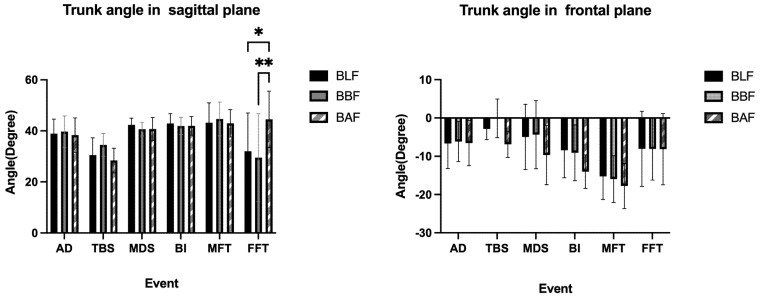
Trunk angles in sagittal and frontal planes. Comparisons of BLF, BBF, and BAF during AD, TBS, MDS, BI, MFT, and FFT (For sagittal plane, + stands for flexion, − stands for extension; For frontal plane, + stands for left lateral bending, − stands for right lateral bending). * *p* < 0.05, ** *p* < 0.01.

**Figure 6 ijerph-20-01176-f006:**
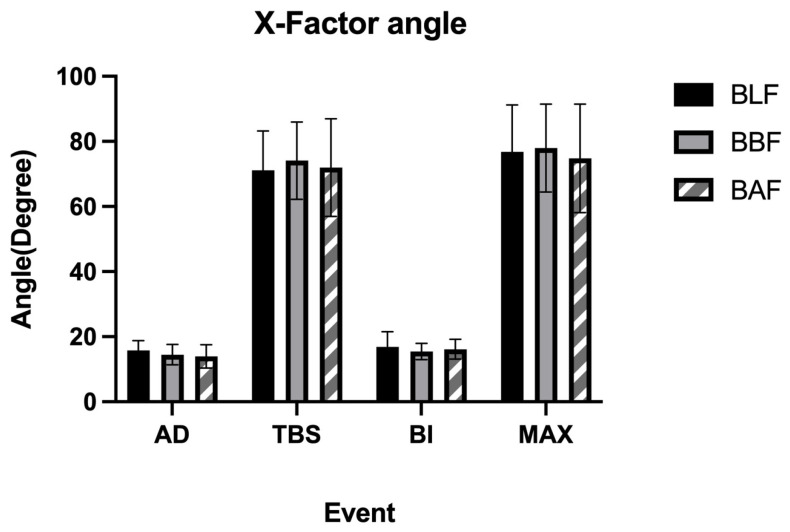
X-factor values at critical events. Comparisons of BLF, BBF, and BAF during AD, TBS, BI, and MAX.

**Table 1 ijerph-20-01176-t001:** Subject characteristics (N = 8).

Characteristics	Age (yr)	Height (cm)	Body Mass (kg)	Handicap	Experience (yr)
Mean ± SD	50.67 ± 9.3	171.5 ± 4.5	69.8 ± 6.3	18.8 ± 1.2	5.2 ± 3.9

Note: the handicap calculation in this study represents the average level of the golfers in the last three rounds of 18-hole play according to the USGA handicap system.

**Table 2 ijerph-20-01176-t002:** EMG (MVC) by Lead-ES(T8) and Trail-ES(T8) muscle results for the BLF, BBF, and BAF at different events.

Phase	Slope	Lead-ES (T8)			Trail-ES (T8)		
		M ± SD	MD (95%CI)		M ± SD	MD (95%CI)	
			a	b		a	b
BS	a.BLF	13.74 ± 2.47			21.58 ± 3.86		
	b.BBF	12.28 ± 2.88	1.45 (−2.66, 5.57)		19.17 ± 5.60	2.41 (−1.22, 6.04)	
	c.BAF	15.79 ± 4.48	−2.06 (−6.17, 2.06)	−3.51 (−7.62, 0.61)	19.39 ± 4.77	2.19 (−1.44, 5.82)	−0.22 (−3.85, 3.41)
FS	a.BLF	21.58 ± 2.81			24.65 ± 3.67		
	b.BBF	20.10 ± 3.89	1.47 (−2.64, 5.59)		21.31 ± 4.28	3.34 (−0.29, 6.97)	
	c.BAF	23.24 ± 5.12	−1.66 (−5.78, 2.46)	−3.14 (−7.25, 0.98)	22.63 ± 3.40	2.01 (−1.62, 5.64)	−1.32 (−4.95, 2.31)
ACC	a.BLF	39.21 ± 4.86			33.97 ± 3.37		
	b.BBF	36.78 ± 4.27	2.43 (−1.68, 6.55)		36.19 ± 4.47	−2.22 (−5.85, 1.41)	
	c.BAF	35.70 ± 3.35	3.51 (−0.61, 7.63)	1.08 (−3.04, 5.19)	33.69 ± 4.24	0.28 (−3.35, 3.91)	2.50 (−1.13, 6.13)
EFT	a.BLF	51.81 ± 3.86			27.96 ± 3.78		
	b.BBF	42.73 ± 3.86	9.08 **** (4.97, 13.2)		27.28 ± 2.46	0.68 (−2.95, 4.31)	
	c.BAF	44.70 ± 5.06	7.11 *** (3.00, 11.23)	−1.97 (−6.08, 2.15)	25.41 ± 2.76	2.55 (−1.09, 6.18)	1.87 (−1.76, 5.50)
LFT	a.BLF	30.85 ± 3.83			19.49 ± 4.29		
	b.BBF	25.62 ± 3.35	5.23 ** (1.11, 9.34)		17.16 ± 3.71	2.33 (−1.30, 5.96)	
	c.BAF	25.74 ± 2.80	5.11 * (0.99, 9.23)	−0.12 (−4.23, 4.00)	16.21 ± 3.67	3.28 (−0.35, 6.92)	0.95 (−2.68, 4.58)

Note: BLF: a, BBF: b, BAF: c. MD: mean difference. Differences in comparison are indicated by * *p* < 0.05, ** *p* < 0.01, *** *p* < 0.001, **** *p* < 0.0001.

**Table 3 ijerph-20-01176-t003:** EMG (MVC) by Lead-ES(L3) and Trail-ES(L3) muscle results for the BLF, BBF, and BAF at different events.

Phase	Slope	Lead-ES (L3)			Trail-ES (L3)		
		M ± SD	MD (95%CI)		M ± SD	MD (95%CI)	
			a	b		a	b
BS	a.BLF	17.12 ± 4.33			12.56 ± 2.92		
	b.BBF	17.27 ± 4.55	−0.15 (−3.26, 2.96)		12.77 ± 2.89	−0.21 (−4.38, 3.95)	
	c.BAF	16.97 ± 4.89	0.15 (−2.96, 3.26)	0.30 (−2.81, 3.41)	12.27 ± 2.52	0.29 (−3.88, 4.46)	0.50 (−3.67, 4.67)
FS	a.BLF	24.42 ± 4.11			22.12 ± 3.23		
	b.BBF	25.68 ± 3.32	−1.26 (−4.37, 1.85)		23.09 ± 3.88	−0.97 (−5.14, 3.20)	
	c.BAF	24.91 ± 3.26	−0.49 (−3.60, 2.62)	0.77 (−2.34, 3.88)	19.27 ± 3.34	2.85 (−1.32, 7.02)	3.82 (−0.35 7.99)
ACC	a.BLF	48.84 ± 4.07			46.85 ± 3.35		
	b.BBF	45.56 ± 3.75	3.28 * (0.17, 6.39)		42.98 ± 3.35	3.87 (−0.30, 8.04)	
	c.BAF	45.30 ± 4.07	3.54 * (0.43, 6.64)	0.25 (−2.86, 3.36)	49.00 ± 4.94	−2.15 (−6.32, 2.02)	−6.02 * (−10.19, −1.85)
EFT	a.BLF	39.33 ± 3.98			38.14 ± 4.88		
	b.BBF	34.86 ± 4.55	4.47 ** (1.36, 7.58)		35.16 ± 4.79	2.98 (−1.19, 7.15)	
	c.BAF	30.40 ± 4.93	8.93 **** (5.82, 12.04)	4.46 ** (1.35, 7.57)	37.28 ± 4.71	0.86 (−3.31, 5.02)	−2.13 (−6.29, 2.04)
LFT	a.BLF	22.43 ± 3.20			26.76 ± 4.54		
	b.BBF	22.17 ± 2.96	0.26 (−2.85, 3.37)		24.39 ± 4.54	2.37 (−1.80, 6.54)	
	c.BAF	24.31 ± 3.99	−1.88 (−4.99, 1.23)	−2.14 (−5.25, 0.97)	21.43 ± 4.05	5.33 * (1.16, 9.50)	2.96 (−1.21, 7.13)

Note: BLF: a, BBF: b, BAF: c. MD: mean difference. Differences in comparison are indicated by * *p* < 0.05, ** *p* < 0.01, **** *p* < 0.0001.

**Table 4 ijerph-20-01176-t004:** Trunk angle in sagittal results for the BLF, BBF, and BAF at different events.

Event	Slope	M ± SD	MD (95%CI)	
			a	b
AD	a.BLF	38.92 ± 5.71		
	b.BBF	39.73 ± 6.15	−0.81 (−11.59, 9.97)	
	c.BAF	38.36 ± 6.02	0.56 (−10.22,11.34)	1.38 (−9.40, 12.15)
TBS	a.BLF	30.54 ± 6.74		
	b.BBF	34.50 ± 4.46	−3.96 (−14.74, 6.81)	
	c.BAF	28.40 ± 4.30	2.14 (−8.64, 12.92)	6.10 (−4.68, 16.88)
MDS	a.BLF	42.36 ± 2.59		
	b.BBF	40.67 ± 2.65	1.69 (−9.09, 12.47)	
	c.BAF	40.74 ± 4.01	1.62 (−9.16, 12.40)	−0.07 (−10.85, 10.71)
BI	a.BLF	42.91 ± 3.83		
	b.BBF	41.95 ± 3.39	0.97 (−9.81, 11.75)	
	c.BAF	41.98 ± 3.27	0.93 (−9.84, 11.71)	−0.03 (−9.11, 12.44)
MFT	a.BLF	43.23 ± 7.73		
	b.BBF	44.65 ± 5.93	−1.42 (−12.20,9.36)	
	c.BAF	42.99 ± 4.79	0.24 (−10.53, 11.02)	1.66 (−9.11, 12.44)
FFT	a.BLF	32.00 ± 15.04		
	b.BBF	29.52 ± 15.37	2.48 (−8.29, 13.26)	
	c.BAF	44.53 ± 9.91	−12.52 * (−23.30, −1.75)	−15.01 ** (−25.78, −4.23)

Note: BLF: a, BBF: b, BAF: c. MD: mean difference. Differences in comparison are indicated by * *p* < 0.05, ** *p* < 0.01.

**Table 5 ijerph-20-01176-t005:** Trunk angle in frontal results for the BLF, BBF, and BAF at different events.

Event	Slope	M ± SD	MD (95%CI)	
			a	b
AD	a.BLF	−6.64 ± 6.56		
	b.BBF	−6.15 ± 5.23	−0.49 (−10.13, 9.16)	
	c.BAF	−6.54 ± 5.29	−0.10 (−9.74, 9.54)	0.39 (−9.26, 10.03)
TBS	a.BLF	−2.83 ± 2.82		
	b.BBF	−0.10 ± 5.05	−2.73 (−12.38, 6.91)	
	c.BAF	−6.86 ± 3.09	4.02 (−5.62, 13.67)	6.76 (−2.89, 16.40)
MDS	a.BLF	−4.95 ± 8.51		
	b.BBF	−4.36 ± 8.88	−0.58 (−10.23, 9.06)	
	c.BAF	−9.69 ± 6.91	4.74 (−4.90, 14.38)	5.32 (−4.32, 14.97)
BI	a.BLF	−8.41 ± 7.21		
	b.BBF	−9.07 ± 7.30	0.66 (−8.98, 10.31)	
	c.BAF	−14.01 ± 3.92	5.60 (−4.04, 15.25)	4.94 (−4.70, 14.58)
MFT	a.BLF	−15.24 ± 6.06		
	b.BBF	−15.95 ± 5.47	0.71 (−8.93, 10.35)	
	c.BAF	−17.77 ± 5.26	2.52 (−7.12, 12.17)	1.81 (−7.83,11.46)
FFT	a.BLF	−8.06 ± 9.81		
	b.BBF	−8.07 ± 7.29	0.01 (−9.63, 9.65)	
	c.BAF	−8.14 ± 8.31	0.08 (−9.57, 9.72)	0.07 (−9.58, 9.71)

Note: BLF: a, BBF: b, BAF: c. MD: mean difference.

**Table 6 ijerph-20-01176-t006:** X-factor angle results for the BLF, BBF, and BAF at different events.

Event	Slope	M ± SD	MD (95%CI)	
			a	b
AD	a.BLF	14.49 ± 3.16		
	b.BBF	15.83 ± 2.98	−1.34 (−6.39, 3.72)	
	c.BAF	13.97 ± 3.59	0.52 (−4.54, 5.58)	1.86 (−3.20, 6.92)
TBS	a.BLF	74.09 ± 11.86		
	b.BBF	71.15 ± 12.08	2.94 (−17.36, 23.24)	
	c.BAF	71.98 ± 14.99	2.11 (−18.20, 22.41)	−0.83 (−21.13, 19.47)
BI	a.BLF	15.45 ± 2.48		
	b.BBF	16.85 ± 4.70	−1.41 (−6.89, 4.08)	
	c.BAF	16.16 ± 3.01	−0.71 (−6.20, 4.77)	0.69 (−4.79, 6.18)
MAX	a.BLF	77.97 ± 13.52		
	b.BBF	76.76 ± 14.46	1.21 (−22.03, 24.44)	
	c.BAF	74.78 ± 16.66	3.19 (−20.04, 26.42)	1.98 (−21.25, 25.22)

Note: BLF: a, BBF: b, BAF: c. MD: Mean difference.

**Table 7 ijerph-20-01176-t007:** Hit Parameter Results for BLF, BBF, and BAF measured by launch monitor (mean ± SD).

Variable	Slope	M ± SD	MD (95%CI)	
			a	b
Ball speed (mph)	a.BLF	103.24 ± 4.88		
	b.BBF	97.49 ± 8.08	5.75 ** (1.87, 9.63)	
	c.BAF	92.99 ± 6.75	8.93 **** (4.96, 12.90)	3.18 (−0.79, 7.15)
Swing speed (mph)	a.BLF	75.90 ± 4.39		
	b.BBF	72.26 ± 4.32	3.63 * (0.88, 6.39)	
	c.BAF	70.39 ± 4.43	4.88 *** (2.06, 7.70)	1.25 (−1.57, 4.07)
Carry (yards)	a.BLF	141.26 ± 8.31		
	b.BBF	131.18 ± 15.60	10.08 ** (2.65, 17.51)	
	c.BAF	123.35 ± 13.82	15.41 **** (7.82, 23.01)	5.33 (−2.26, 12.93)
Smash factor	a.BLF	1.36 ± 0.05		
	b.BBF	1.35 ± 0.06	0.01 (−0.03, 0.06)	
	c.BAF	1.32 ± 0.08	0.03 (−0.02, 0.08)	0.02 (−0.03, 0.07)
Launch angle (degrees)	a.BLF	20.66 ± 2.51		
	b.BBF	20.16 ± 1.63	0.50 (−1.30, 2.30)	
	c.BAF	19.63 ± 1.62	1.36 (−0.48, 3.20)	0.86 (−0.98, 2.70)
Apex (ft)	a.BLF	25.33 ± 5.76		
	b.BBF	21.49 ± 5.99	3.84 * (0.01, 7.67)	
	c.BAF	17.98 ± 2.05	7.05 ** (3.13, 10.96)	3.21 (−0.71, 7.13)

Note: BLF: a, BBF: b, BAF: c. MD: Mean difference. Differences in comparison are indicated by * *p* < 0.05, ** *p* < 0.01, *** *p* < 0.001, **** *p* < 0.0001.

## Data Availability

The data presented in this study are available on request from the corresponding author.

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
