# Peer review of "Effects of Ground Slopes on Erector Spinae Muscle Activities and Characteristics of Golf Swing"

_ijerph, 2023, doi:10.3390/ijerph20021176_

Round 1

Reviewer 1 Report

I would like to congratulate the authors for this work on a little known sport such as golf. Despite the relevance and importance of the work, there are some aspects that I would like to see modified. 

The introduction is very well written, indicating the aspects that are relevant in golf. only a few doubts remain. For example, does the X-factor influence the production of injuries (increasing or decreasing it), as well as other technical aspects? only the usefulness of this aspect in hitting has been reported.

In the introduction, lines 94-97, it is specified that other variables have been analyzed, but it is not clear if this is an objective of the research. If it is, include an objective specifying that these variables have been analyzed according to the different slopes, as well as the hypothesis to be contrasted. 

In the participants section, what do the authors consider to be a healthy golfer? And why only right-handers? Was it a chosen aspect or was it because there were only right-handed golfers in the sample? 

There is a lack of information in the method section. For example, in what order did the participants strike? Was it random? How much rest time was left between each stroke? Why? Was the hitting procedure randomized, or was it similar for all (e.g., everyone did the non-slope first, then BAF, then BBF)?

In the results section, the graphs are very representative, but only include the p-value. It would be nice to know the effect size, confidence intervals, among other aspects, at least for the pairs that show statistically significant differences. Therefore, please complete the information requested at the bottom of the graphs or by including it in the text. 

Similarly, in Table 2, the ICCs and effect size would be missing. 

In the discussion section, it would be good to find a first paragraph stating the objective(s) of the study, as well as the hypotheses proposed, since they make it easier for the reader to follow the thread. 

As stated in the objectives, a second objective regarding the ball parameters is needed and should also be included in the discussion section and should include the hypothesis that is put forward for this objective. 

The conclusions must respond to the stated objectives, especially if a second objective is included with respect to the parameters of the ball. 

Minor issues

- In the abstract, line 18, "3)" is missing. The sentence would read: and 3) a slope....

- In the introduction, the first time BAF and BBF appear, specify what these acronyms stand for since you have only previously specified them in the abstract. 

Author Response

Dear reviewer,

Thanks for your comments. Please check the revised version.
Thank you.

Reviewer 2 Report

Dear Authors,

It is my pleasure to review your study but I have a few doubts.

General information:

-The article should be prepared in accordance with the journal's guidelines.

-The abstract should be divided into sections (background, methods, result etc.). It is then clearer to the reader.

-References should be prepare in accordance with the guidelines. It should be corrected.

-Do not repeat the words in the title in key words.

Introduction:

-The introduction is clearly presented but the aim of the study  should be added.

M&M:

- n=8, what was the sample size value?

-Figure 1. is of poor quality,

-what is the measurement error in the OptiTrack Motion Capture System?

-Figure 3. is also poor quality,

Result:

-Figure 4 presents statically significant results, but is no data/values,

-the same situation is in 3.3 Comparison of trunk angle and X-factor,

Discusion:

-the study is on a small group of participants, please added to limitation section

-for scientific quality, you please add more newer references to the discussion.

Author Response

Dear reviewer,

Thanks for your comments. Please check that the revised version has been uploaded.
Thank you.

Reviewer 3 Report

I find the manuscript novel and well written. The diagrams are very helpful.

My only comment is about the "handicap" presented in table 1. What is this a measure of? I assume it represents previous golf round scores. This can be confusing to the reader. It requires clarification

Author Response

Dear reviewer,

Thanks for your comments. Please check the revised version.

Reviewer 4 Report

Please see the file.

Author Response

(The authors gave the same response as above.)

Author Response

(The authors gave the same response as above.)

Reviewer 6 Report

The article is interesting and raises the following doubts.

Line 15-16 and 33-35

The study purpose (ES EMG and Kinematics), study design (ES EMG and Kinematics), and conclusions (EMG) should remain linked and consistent.

Line 162-165           

The purpose of this study seems to be primarily related to back injuries, is it appropriate to calculate trunk angles with the thoracic spine relative to the laboratory in the study methodology?

Line 220

Figure 5 Trunk frontal plane angles are indicated, with positive and negative indication (left and right lateral flexion).

Recommendation.

Horizontal plane motion in golf seems to be very important for back pain, add trunk motion in the horizontal plane in the results of the study in Figure 5 section.

Line227

Figure 6 positive and negative to represent the x-factor (shoulder hip separation angle), easier to understand the relationship between clockwise and counterclockwise changes.

The change in trunk angle (+/-) makes it easier to understand the passive role of the erector spinae during the golf swing, possibly 1, as a soft tissue. 2, centripetal movement. 3, centrifugal control. 4, stabilizing action.

Line 259-263

Please rephrase it to make logics clearer.

Line 285-286 and 288-291
Please rephrase it and cite the references.

Author Response

(The authors gave the same response as above.)

Reviewer 7 Report

I think it's a very interesting study for the reader. However, the author needed some minor corrections and grammar fixation. Please re-check paragraphs.

Introduction

1.     There are some sentences that are need to be fixed.

A.     According to Peter et al. [2], approximately 80% of the swings on most golf courses are conducted on a slope of plus or minus 10°, which could be an uphill slope, downhill 45 slope, BAF slope, or BBF slope [3,4].

B.      The slope in a game of golf may be affecting golfers’ posture at the address. A golfer 49 should adjust the center line of the body to adapt to the change in the slope [5].

C.      When a motion is introduced in one plane, it 52 modifies motions in the other two planes

Method

1.     range is not required information. SD is enough.

Results

1.     show result tables that include mean, SD, and statistical result of the angle data.

2.     make this explanation in sentence.

A.     BLF 49% MVCBBF45% MVC, BAF 45% MVC

3.     In <table 2>, Swing speed and smash factor’s p value is both 0.05. And one is significant, and the other is not. Please make this clear.

Discussion

1.     In line 359, to support the paragraph, the author need to show the related results of the research.

Author Response

(The authors gave the same response as above.)

Round 2

Reviewer 1 Report

The authors have resolved all the indications of the first review.

Reviewer 2 Report

Dear Authors,

the article looks much better.

Thank you for making the corrections.

I have no objections.

Best regards